# Pain in Long-Term Cancer Survivors: Prevalence and Impact in a Cohort Composed Mostly of Breast Cancer Survivors

**DOI:** 10.3390/cancers16081581

**Published:** 2024-04-20

**Authors:** Concepción Pérez, Dolores Ochoa, Noelia Sánchez, Ana Isabel Ballesteros, Sheila Santidrián, Isabel López, Rebeca Mondéjar, Thiago Carnaval, Jesús Villoria, Ramón Colomer

**Affiliations:** 1Pain Clinic, Hospital de La Princesa, 28006 Madrid, Spain; dochoa@iis-princesa.org (D.O.); noe.sm@live.com (N.S.); sheilasanber@gmail.com (S.S.); isable1977@hotmail.com (I.L.); 2Department of Clinical Oncology, Hospital de la Princesa, 28006 Madrid, Spain; anabelballes@gmail.com (A.I.B.); rmondejar@hotmail.com (R.M.); ramon.colomer@salud.madrid.org (R.C.); 3Department of Design and Biometrics, Medicxact, S.L., 28430 Alpedrete, Spain; carnavalthiago@medicxact.es (T.C.); villoriajesus@medicxact.es (J.V.)

**Keywords:** long-term cancer survivors, neuropathic pain, breast cancer, quality of life, patient outcome assessment

## Abstract

**Simple Summary:**

Although cancer survival is increasing and many survivors report having to endure pain, its prevalence and consequences have received little attention. We found that pain is associated with reduced quality of life as well as diminished emotional and professional performance in long-term cancer survivors. In addition, a neuropathic component often goes underdiagnosed and/or undertreated. Adopting appropriate neuropathic pain diagnostic tools should be standard clinical practice and targeting the neuropathic component seems to be a good, yet underused therapeutic approach which has the potential to improve cancer survivors’ health outcomes.

**Abstract:**

Cancer survival is becoming more common which means that there is now a growing population of cancer survivors, in whom pain may be common. However, its prevalence has hardly been addressed systematically. We aimed to assess the prevalence and explore the pathophysiology and impact of pain on health outcomes in cancer survivors. We conducted a retrospective–prospective cohort study in cancer-free patients diagnosed with cancer at least five years before the study start date. We used multivariable regression to establish the association of patients’ cancer characteristics with pain, and then the association of patients’ pain features with health outcomes and related symptoms. Between March and July 2021, 278 long-term cancer survivors were evaluated. Almost half of them (130/278, 46.8%) had pain, of whom 58.9% had a probable neuropathic component, but only 18 (13.8%) were taking specific drugs for neuropathic pain. A history of surgery-related pain syndrome in breast cancer patients was more than twice as frequent in the pain cohort. Post-chemotherapy and post-radiotherapy pain syndromes were uncommon. Pain was associated with lower QoL, emotional functioning, professional performance, and disability scores. Pain is a frequent health determinant in cancer survivors. Referral to specialised pain services may be a reasonable move in some cases.

## 1. Introduction

Pain is the most common symptom of cancer at diagnosis and rises in prevalence throughout and beyond cancer treatment [1]. Cancer survival has considerably increased in recent years due to major advances in diagnosis, treatment, and rehabilitation. The 5-year cancer survival rate in Europe exceeds 60% for the most common tumours [2], and around 40% of patients are alive over 10 years after diagnosis [3]. Each year, this should account for over 100,000 new long-term survivors in a medium-sized Western country like Spain [4], many of whom will fully overcome the oncologic disease but will continue to be in pain [1]. The impact of pain is compounded in these patients by special physical, social, occupational, psychological, and emotional needs [4].

Causes of cancer-related pain include the tumour itself or its metastases inflaming or eroding bone, viscera, or nerves, or pain related to tissue or nerve damage induced by cancer treatments [1,5,6]. Surgery is key in the treatment, diagnosis, and palliation of cancer. Chronic post-surgical pain has been described after different procedures (e.g., thoracotomy, breast surgery, modified radical neck dissection), and the common mechanism is central sensitization. It typically presents 2–12 months after surgery, although some patients might experience neuropathic symptoms immediately after or even years later [7]. Incidences can reach 80% for phantom limb pain [7] or chronic post-thoracotomy pain [8]. Chronic post-chemotherapy (CT) pain can be an important drawback for antineoplastic agents since many are neurotoxic. CT neurotoxicity may affect the central nervous system, but CT-induced peripheral sensory neuropathy is more prevalent, affecting from a few to as much as 100% of exposed patients depending on the patients’ comorbidities, the type of CT used, and the cumulative dose received [9]. CT-induced peripheral neuropathy can often be severe enough to require dose adjustments or CT cessation, leading to potentially suboptimal therapy [10]. Likewise, radiotherapy (RT) causes very common painful acute side effects such as dermatitis and, particularly, mucositis, but can also result in several potentially painful conditions that can manifest months or even years after treatment, including osteoradionecrosis, plexopathies and pelvic pain syndrome [7]. Estimates hover around 5–9% for RT-specific brachial plexopathy (more common with higher doses of radiation) [11], and 10–15% for chronic pelvic syndrome [12].

Our objectives were to estimate the prevalence of pain in general, the major cancer pain syndromes, and associated symptoms in long-term cancer survivors. We also attempted to delve into the pathophysiology and predisposing factors for pain in these patients and assess the impact on their quality of life (QoL), emotional functioning, professional performance, and disability.

## 2. Materials and Methods

### 2.1. Study Design and Setting

We conducted a single-centre, retrospective–prospective cohort study at La Princesa University Hospital, Madrid, Spain. Patients were recruited from the Medical Oncology outpatient clinic, which receives about 5.0% of nearly 361,000 follow-up outpatient consultations carried out annually in the hospital (most patients come several times each year). The period evaluated preceded the data collection date because the influence of cancer on outcomes was evaluated together with that of pain (see the statistical methods). It was designed and overseen by the investigators, who are listed as authors. The Ethics Committee of La Princesa University Hospital approved the study protocol prior to starting. It was performed in accordance with local regulations, Good Clinical Practice Guidelines, the principles of the Declaration of Helsinki, and current regulations governing the protection of personal data and the rights and responsibilities concerning information and documentation in healthcare.

### 2.2. Study Subjects

Cancer-free patients diagnosed with cancer at least five years prior to the study start date attending for routine follow-up visits during the inclusion period, and who met all the inclusion criteria and none of the exclusion criteria, were selected for the study. Patients included were men and women aged 18 years or older, diagnosed with cancer at least 5 years before inclusion, currently in disease-free survival, and who signed the written informed consent. We excluded patients currently fighting active cancer despite radical treatment, patients unable/unwilling to answer the study questionnaires, patients unfit to undergo all study procedures, patients diagnosed with psychiatric or neurological disorders that could affect their participation, or patients under pharmacological treatments that could interfere with the ability to understand or answer the questionnaires.

### 2.3. Clinical Endpoints

The primary endpoint was the number and percentage of long-term survivors reporting pain among the included population. Secondary endpoints included the following: (a) features of cancer disease and therapy, (b) the intensity and interference of general pain and of each major oncological pain syndrome measured with the numerical rating scales of the Brief Pain Inventory (BPI), which range from 0 (no pain) to 10 (the worst imaginable pain) [13], (c) the prevalence of pain labelled as nociceptive, neuropathic, or mixed when the *Douleur Neuropathique 4 Questions* (DN4) instrument was scored at 4 or over 10 [14], (d) the prevalence of each level of the 5 dimensions assessed by the EuroQoL 5 Dimensions (EQ5D) questionnaire and the QoL scores provided by this instrument [15], (e) the prevalence of borderline or pathological anxiety, depression, or catastrophism according to the Hospital Anxiety and Depression Scale (HADS) [16] and the Pain Catastrophising Scale (PCS) [17], (f) the scores of the dimensions of the Work Productivity and Activity Impairment (WPAI) Pain and General Health questionnaires, including disability [18], and (g) the prevalence of pain associated symptoms (insomnia and fatigue). See the footnotes of Table 3 for more details about scoring ranges and interpretation. Data on oncological disease were collected retrospectively. Data on pain prevalence and consequences were collected prospectively.

### 2.4. Data Sources

Medical oncologists recruited potential candidates in their outpatient clinics. Accepted participants were subsequently referred to individual interviews with members of the study team who performed the study-specific visits and searched the medical files for additional information.

### 2.5. Study Size

Given that this was a single-centre study, we did not perform formal power-based sample size calculations. Nonetheless, we expected to recruit 300 patients within a reasonable time period, with whom we could achieve a sufficient precision of about ±5% in the most unfavourable case that the prevalence was 50%. These calculations have been conducted with PASS Power Analysis and Sample Size Software (2021 version), NCSS, LLC, Kaysville, Utah, 84037 USA (see ncss.com/software/pass—last accessed on 11 April 2024 for technical details, including the algebraic expressions used).

### 2.6. Statistical Methods

Analyses were performed on a locked database. This database was created ad hoc by the study investigators, who were also responsible for data entry and integrity. Prior to analysis, we performed database consistency and integrity checks.

All analyses were carried out on the available data. We performed standard descriptive analyses of all collected data. Categorical variables were expressed as absolute and relative frequencies. The variables obtained from scores, scales, and questionnaires were expressed as proportions (prevalence) and means (standard deviations, SD), as applicable. Inferences to the source population were performed by means of the 95% exact or asymptotic confidence intervals (CI). Bivariable inferences between patients with and without pain were performed using either *t*/Mann–Whitney’s U or Fisher’s exact/Pearson’s chi-square tests, as appropriate.

We used multivariable regression to first establish the association of patients’ characteristics and cancer characteristics (sociodemographic data, medical history, cancer stage, cancer therapies, etc.) with the pain in order to assess the predisposing factors. Secondly, we studied the association of patients’ characteristics and their pain features with health outcomes (QoL, emotional functioning, work productivity, and disability) and related symptoms to assess the impact of cancer and pain on them. These regression models were further used to obtain estimates of the distinct and joint causal effects of cancer and pain on health outcomes via mediation analyses, the results of which will be published in a separate article.

All analyses were performed with the *SAS* statistical software, version 9.4 (SAS Institute, Cary, NC, 27607 USA).

## 3. Results

### 3.1. Patient Disposition, Pain Prevalence, Cancer, and Pain Characteristics

Between March and July 2021, of 287 long-term cancer survivors ascertained from Medical Oncology consultations, 278 (96.9%) agreed to participate. Their overall mean (SD) age and body mass index were 63.7 (11.5) years old and 26.4 (4.6) kgm^−2^, respectively, and the vast majority were Caucasian women. Over 50% of patients were married and almost half (136/277, 49.1%) were retired. Less than 40% had had access to higher education (i.e., university degree) (Table 1). One hundred and thirty (46.8%) suffered pain (Figure 1). The interval estimation (95% CI) placed this prevalence in the population between 40.8% and 52.8%. This is consistent with the prevalence of pain observed overall among patients attending the outpatient clinics of our hospital (~45.3%, internal data held by the authors). According to the DN4, the pain had a probable neuropathic component in 76 of 129 (58.9%) pain patients (27.3% of the whole sample). There were some differences between patients with and without pain. Patients in pain had a significantly higher mean body mass index, were more frequently women, and had more chronic painful conditions, whilst paid jobs were more common among pain-free patients (Table 1).

The mean (SD) overall time elapsed since the first tumour diagnosis was 11.6 (5.6) years when patients had a mean (SD) age of 52.1 (12.1) years. This was evenly distributed between cohorts (Table 2). Although most patients had just one primary tumour, the proportion of those having more than one tumour was significantly greater in the pain cohort. Breast cancer was, by far, the most common primary tumour, followed by colon cancer. The frequencies of the types of tumours did not differ significantly between patients with and without pain. Nearly all patients had undergone surgical treatments, but the number of surgeries was higher among patients with pain (Table 2). More than 80% of patients received chemotherapy and about one-third received radiotherapy. Whilst the proportion of patients who received chemotherapy did not differ between cohorts, radiotherapy was significantly more common among patients with pain (Table 2). The most frequent drug classes delivered in chemotherapy were alkylating agents, followed by antitumor antibiotics and antimetabolites. The distribution between patients with and without pain was uneven: alkylating agents, antitumor antibiotics, and aromatase inhibitors were significantly more common in patients with pain, whilst antimetabolites and platinum analogues were so in pain-free patients (Table 2 and Appendix A). Of the agents considered as potentially neurotoxic, only taxanes were relatively common, but their use was not significantly different between pain and pain-free patients (Table 2). The majority of patients did not have advanced disease at the time of diagnosis, T_1_N_0_M_0_ being the most frequent (21.3%) staging level. Other common TNM stages found were T_2_N_0_M_0_ (14.5%), T_1_N_1_M_0_ (11.1%), and T_2_N_1_M_0_ (8.1%). The proportion of patients from each staging was similar in both cohorts. Markedly, TNM staging could not be retrieved for 43 patients. For details about Tumour, Node, and Metastasis (TNM) cancer staging, see the publication by the National Cancer Institute at https://www.cancer.gov/about-cancer/diagnosis-staging/staging (accessed on 11 April 2024).

Of the 130 patients in the pain cohort, 99 (76.2%) were taking at least one kind of analgesic drug. The most frequently used drug classes were (multiple choices possible) anilides/acetaminophen (46/99 patients, 46.5%), the nonsteroidal anti-inflammatory propionic acid derivatives, opioids (19/99 patients, 19.2%, each), antiepileptic drugs (15/99 patients, 15.2%), and pyrazolines (13/99 patients, 13.1%). These can be classified into the analgesic ladder steps 1 (81/99 patients, 81.8%), 2 (19/99 patients, 19.2%), and 3 (2/99 patients, 2.0%; all 19 patients on opioids were receiving tramadol, but just 2 were also on transdermal fentanyl). A total of 18 out of 99 patients (18.2%, 13.8% of the pain cohort) were on specific drugs for neuropathic pain (antiepileptics: 15 patients, topical capsaicin: 2 patients, and triptans: 1 patient).

### 3.2. Clinical Endpoints, Pain Characteristics and Health Outcomes

The most common pain sites were the upper and lower extremities and the neuroaxis. Mean (SD) pain intensities were 4.0 (2.3) for the average pain intensity in the last 24 h, 5.5 (2.5) and 2.0 (2.0), respectively, for the maximum and minimum pain intensities in the last 24 h, and 4.6 (2.3) for the current pain intensity. Appendix A shows the scores’ distribution by each intensity category.

A history of post-surgical pain syndromes was not a frequent finding in either cohort, except for the breast cancer surgery-related pain syndrome, which was more than twice as frequent in the pain cohort (43.0% vs. 19.3% among pain-free patients). Both post-CT and post-RT pain syndromes were uncommonly reported in either cohort.

Pain on the sides of the thorax and limbs was significantly more frequent in the probable neuropathic pain subgroup, and pain in the neuroaxis was more frequent in the probable nociceptive pain subgroup (Appendix A). In addition to age and body mass index (BMI), having had three or more surgeries was significantly associated with neuropathic pain features; chemo and radiotherapies were not (Appendix A). However, this does not imply that neuropathic pain was absent in patients who underwent chemo or radiotherapy, but rather that it was equally frequent in those who did not.

Pain patients had more disabilities and emotional distress in the EQ5D categories than pain-free patients (Appendix A). Both the QoL Index and the EQ5D Health Status score were statistically significantly lower (worse) in the pain cohort than in the pain-free cohort (Mann–Whitney *p* < 0.001, Table 3, Appendix A). Likewise, borderline and abnormally high anxiety and depression scores in the HADS were more than five times more frequent in pain patients than in pain-free patients (Appendix A). The scores of these subscales were statistically significantly higher (worse) in the pain cohort than in the pain-free cohort (Mann–Whitney *p* < 0.001, Table 3, Appendix A). The PCS scores were also higher (worse) in the pain cohort than in the pain-free cohort (Appendix A), but this is of little relevance given that the PCS constructs are not pertinent for pain-free patients.

Significantly more patients had a paid job in the pain-free cohort (51/148, 34.5%) than in the pain cohort (29/130, 22.3%; Pearson’s chi-square *p*-value: 0.026). Disability scores of the WPAI General Health and Pain Scales were significantly higher (worse) in the pain cohort than in the pain-free cohort (Table 3, Appendix A). The proportions of patients suffering insomnia or fatigue were also significantly greater in the pain cohort (Table 3).

### 3.3. Adjusted Analyses

The multivariable regressions showed that pain had significant direct associations with BMI, previous chronic diseases, surgeries and treatment with aromatase inhibitors, and significant inverse associations with male gender and breast cancer (Table 4). Pain of probable neuropathic origin was significantly associated only with age, BMI, and surgeries (Table 5). In turn, the presence of pain was the single predictor that was consistently and significantly associated with worse outcomes throughout all clinical endpoints except unemployment and presenteeism. The association was stronger for pain of probable neuropathic origin than for nociceptive pain (Appendix A). The association of other patients’ features was variable, with age, BMI, and cancer therapies the most consistently associated with outcomes. With the exception of anxiety, the greater the age and BMI, and the more intense the anti-cancer therapies in general, the worse the outcomes (Appendix A).

## 4. Discussion

This single-centre retrospective–prospective study in a cohort composed mainly of breast cancer survivors found that persisting pain affects almost one-half of the patients. In addition, when present, pain was associated with noticeable impairments of many health outcomes. These impairments were particularly pronounced among patients with probable neuropathic pain. Thus, pain may be a pervasive health determinant in cancer patients once they overcome the oncologic disease.

Pain prevalence was somewhat higher than in a systematic dedicated review of breast cancer patients [19] but similar to a more recent review of persistent post-surgical pain following breast cancer surgery [20]. In fact, nearly all patients in our cohort had undergone some cancer-related surgery. The prevalence of pain with neuropathic features in the latter study (29%) was very similar to ours (27.3%). Prevalence estimations can vary considerably as a result of variations in the definition of cancer survivorship. We followed the European Organisation for Research and Treatment of Cancer Survivorship Task Force convention guidelines which state that qualifying as a survivor requires evidence of no active disease [21]. In this vein, ours would be a conservative approach, as pain prevalence tends to decrease when the oncological disease subsides [22]. Nonetheless, in line with previous research, it seems clear that at least 40% of cancer survivors may have persistent pain [22]. To further compound the situation, this pain seems to include neuropathic features in about half the cases [22,23], these being precisely those who showed the worst outcomes.

It is already accepted that cancer survivors as a whole report poorer health-related quality of life than healthy peers, yet there is considerable heterogeneity and individual determinants remain elusive [24]. Consistent with the still-scant studies on this subject [25], the present research suggests that pain may be key to explaining why some patients have such an impaired quality of life. It is common that cancer survivors who experience pain are more likely to report sleep disorders, mood alteration, and fatigue, which decrease quality of life [6,26]. We also found a strong association between pain, depression, and anxiety. Although pain undoubtedly can produce emotional distress, the latter can also influence the pain experience involving cognitive factors related to the appraisal of the likely cause and perceived vulnerability to cancer returning [27]. Emotional distress in cancer survivors has been even less studied than quality of life, but an upsetting picture of the pervasive deleterious effect pain has on a wide range of health outcomes emerges from our results. Analogous to chronic non-cancer pain, a range of cognitive, behavioural, and emotional processes that are further modulated by the patient’s socio-environmental context to render and sustain the integral pain experience may also apply to cancer survivors [28]. Thus, the biopsychosocial model could provide the most appropriate framework for their pain management.

Guidelines’ mention of non-pharmacological therapies for cancer survivors are rare and almost exclusively dedicated to cancer treatment side effects and not specifically to pain [29,30]. In fact, despite the endorsement of the biopsychosocial model for cancer treatment itself, post-cancer pain treatment remains mostly biomedical and based almost exclusively on pharmacological strategies [30]. This is in stark contrast to what is conducted, for example, for cancer-related fatigue [31]. Even long-term pharmacological management of cancer-related pain was not formally considered until recently [32]. This gap is relevant because the biopsychosocial model for the management of non-cancer chronic pain may not be directly transposable to cancer survivors, since a cancer diagnosis impacts the way patients both experience and communicate pain [27,33]. The clearly poor PCS scores in our patients in pain suggest that this issue may well be relevant for them [34], hence the importance of targeting pain-related beliefs. This should be carried out with great care, because at the same time, the risk of recurrence should never be dismissed, and new or changing symptoms warrant prompt careful evaluations, but must be carried out in the least intrusive way possible.

We have also found that pain may be neuropathic in at least half of cancer survivors who, incidentally, were those with the worst outcomes. Remarkably, guidelines for neuropathic pain focus almost exclusively on non-cancer pain [35,36]. While there is something published for cancer survivors [37], it is restricted to the widely known chemotherapy-induced neuropathy. In any case, neuropathic pain did not seem to be associated with chemotherapy but instead with surgery. This could relate to the fact that most patients within this cohort were long-term breast cancer survivors who endured a high incidence of neuropathic pain due to mastectomy, whilst the chemotherapy agents used in this type of cancer usually do not associate neurotoxicity [20,38]. Related to this is the fact that treatment with aromatase inhibitors, commonly given for breast cancer, was also associated with the presence of pain. Strikingly, the pain was possibly neuropathic in more than half of patients treated with these agents (see the footnotes of Table 2). This should alert us because they are usually considered to suffer from arthralgia related to hormone therapy [39], but they are seldom thought to endure neuropathic pain.

In addition, our data suggest that neuropathic pain is underdiagnosed and undertreated in cancer survivors, given that most patients were on Step 1 analgesics [40], and only a few received drugs targeting neuropathic pain. Unlike awareness of chemotherapy-induced neuropathy, which seems widespread, post-surgical neuropathic pain might be a relevant and frequently overlooked health problem in cancer survivors (especially in breast cancer survivors) [41]. Furthermore, many such patients might have a neuropathic component even in the absence of an apparent neural injury [41,42]. Adopting appropriate and simple neuropathic pain diagnostic tools (e.g., DN4) in the routine clinical practice of medical oncologists and defining appropriate treatment and referral strategies, for patients with neuropathic features, to pain specialists seems to be a good and underused therapeutic option which could improve health outcomes [43]. Cancer survivors willingly accept the use of tools to obtain patient-reported outcome measures, which may be of great help after the completion of active anti-cancer therapies [44]. In turn, pain clinics, now available in many tertiary centres, can provide an interdisciplinary range of therapies for pain, including pharmacological and interventional therapies, as well as non-pharmacological therapies such as physio- or psychotherapy [45].

This research has limitations. First, the most common oncological diseases in our sample were various types of breast cancer. This has allowed us to assess some pain states typical of breast cancer survivors, such as persistent post-surgical pain or pain associated with aromatase inhibitors. However, other common painful conditions of cancer survivors such as chemotherapy-induced neuropathies, post-radiotherapy pain or pain associated with hematopoietic stem cell transplantation were underrepresented. Since pain characteristics and impact on health outcomes might differ depending on the various mechanisms involved, we purposefully included a dichotomized indicator (breast vs. non-breast cancer, see the Appendix A) in all adjusted analyses, and we did not find any significant association in any of the models. This favours the transposability of our results to other subpopulations, which might be particularly true for neuropathic pain [46,47]. Second, the observational design prevents us from drawing causal conclusions. Yet, we have also used specific statistical methods for performing causal inferences that will be conveyed in a separate report. Third, since this was a single-centre study, representation is limited.

## 5. Conclusions

We deem it appropriate to conclude that pain prevalence in breast cancer survivors, even when restricted to those free from oncological disease, is high and affects 40% or more. In addition, consistent with previous research, persisting pain is associated with considerable impairment of health status, to the point of becoming a ubiquitous health determinant once patients overcome the oncologic disease. Multimodal therapies already in use for chronic non-cancer pain and cancer-related fatigue, which seem to be underused in this condition, could be of great help to those living beyond cancer.

## Figures and Tables

**Figure 1 cancers-16-01581-f001:**
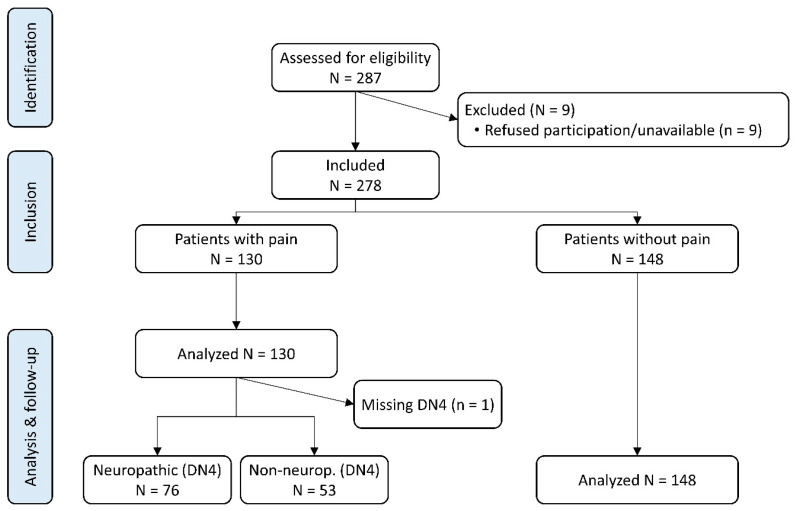
Patient disposition flow diagram.

**Table 1 cancers-16-01581-t001:** Patients’ baseline and demographic characteristics.

	With Pain (*n* = 130)	Without Pain (*n* = 148)	*p*-Value
Age (years), mean (SD)	63.9 (11.3)	63.6 (11.7)	0.975 ^2^
Gender: male, *n* (%)	3 (2.3)	25 (16.9)	<0.001 ^3^
Gender: female, *n* (%)	127 (97.7)	123 (83.1)	
BMI (Kg/m^2^), mean (SD)	27.2 (5.3)	25.7 (3.9)	0.015 ^2^
Race: Caucasian, *n* (%)	118 (90.8)	133 (92.4)	0.663 ^3^
Race: Asian, *n* (%)	1 (0.8)	1 (0.7)	
Race: Latin-American, *n* (%)	9 (6.9)	10 (6.9)	
Race: Middle-Eastern, *n* (%)	2 (1.5)	-	
Marital status: single, *n* (%)	24 (18.5)	30 (20.3)	0.932 ^4^
Marital status: married, *n* (%)	69 (53.1)	76 (51.4)	
Marital status: divorced, *n* (%)	18 (13.9)	17 (11.5)	
Marital status: widow(er), *n* (%)	19 (14.6)	20 (13.5)	
Educational attainment: no studies, *n* (%)	6 (4.7)	7 (5.0)	0.727 ^4^
Educational attainment: secondary, *n* (%)	82 (63.6)	83 (58.9)	
Educational attainment: superior, *n* (%)	41 (31.8)	51 (36.2)	
Employment status: active, *n* (%) ^1^	45 (34.62)	65 (44.22)	0.103 ^4^
Employment status: inactive, *n* (%) ^1^	85 (65.38)	82 (55.78)	
Paid employment, *n* (%)	32 (24.6)	55 (37.2)	0.022 ^4^
Toxic habits: active smoker, *n* (%)	20 (15.38)	22 (14.86)	0.939 ^4^
Toxic habits: active drinker, *n* (%)	13 (10.00)	22 (14.86)	0.207 ^4^
Toxic habits: other, *n* (%)	1 (0.77)	1 (0.68)	>0.999 ^3^
Pain-related associated chronic conditions: any, *n* (%)	59 (57.84)	26 (36.11)	<0.001 ^4^
Pain-related assoc. chronic cond.: osteoporosis, *n* (%)	26 (44.07)	15 (57.69)	0.241 ^4^
Pain-related assoc. chronic cond.: diab. neuropathy, *n* (%)	1 (1.69)	-	>0.999 ^3^
No. of patients per pain type: nociceptive somatic, *n* (%)	129 (99.2)	NA	NA
No. of patients per pain type: nociceptive visceral, *n* (%)	2 (1.5)	NA	
No. of patients per pain type: neuropathic, *n* (%)	57 (43.9)	NA	
No. of patients per pain type: mixed, *n* (%)	57 (44.2)	NA	

Abbreviations: assoc.: associated, BMI: body mass index, cond.: condition, diab.: diabetic, Kg: kilogram, m: metre, No.: number, SD: standard deviation. Relative frequencies were calculated for the population without missing values. ^1^ Active employment status includes patients currently working outside of their households and those working as homemakers; inactive employment status includes retired patients, patients on sick or disability leave, and unemployed patients. ^2^ Mann–Whitney U test. ^3^ Fisher’s exact test. ^4^ Pearson’s chi-square test.

**Table 2 cancers-16-01581-t002:** Disease and therapy characteristics.

	With Pain (*n* = 130)	Without Pain (*n* = 148)	*p*-Value
Elapsed time (years) since tumour diagnosis, mean (SD)	11.8 (5.9)	11.4 (5.2)	0.632 ^5^
Age at cancer diagnosis, mean (SD)	51.9 (12.1)	52.2 (12.2)	0.697 ^5^
Number of reported tumours: one, *n* (%)	109 (83.9)	138 (93.2)	0.013 ^6^
Number of reported tumours: two, *n* (%)	21 (16.2)	10 (6.8)	
Types of primary tumour: inv. duct. breast ca., *n* (%) ^1^	84 (64.6)	78 (52.7)	0.211 ^7^
Types of primary tumour: breast carcinoma, *n* (%) ^1^	7 (5.4)	8 (5.4)	
Types of primary tumour: colon adenocarcinoma, *n* (%) ^1,2^	4 (3.1)	7 (4.7)	
Types of primary tumour: ductal breast carcinoma, *n* (%) ^1,3^	4 (3.1)	6 (4.1)	
Types of primary tumour: inv. lob. breast cancer, *n* (%) ^1,3^	6 (4.6)	4 (2.7)	
Types of primary tumour: colorectal adenocarcinoma, *n* (%) ^1,2^	2 (1.5)	7 (4.7)	
Types of primary tumour: sigmoid adenocarcinoma, *n* (%) ^1,2^	1 (0.8)	4 (2.7)	
Types of primary tumour: nodular melanoma, *n* (%) ^1^	1 (0.8)	3 (2.0)	
Types of primary tumour: invasive breast carcinoma, *n* (%) ^1,3^	-	3 (2.0)	
Types of primary tumour: epidermoid lung carcinoma, *n* (%) ^1^	-	3 (2.0)	
Patients who underwent surgical treatment, *n* (%)	128 (98.5)	145 (98.0)	>0.999 ^7^
No. of surgical interventions required: one, *n* (%)	34 (26.6)	67 (46.2)	0.001 ^6^
No. of surgical interventions required: two, *n* (%)	71 (55.5)	68 (46.9)	
No. of surgical interventions required: three or more, *n* (%)	23 (18.0)	10 (6.9)	
Patients who received CT, *n* (%)	104 (80.0)	127 (85.8)	0.197 ^6^
Number of CT cycles required: one, *n* (%)	10 (9.6)	25 (19.7)	0.118 ^6^
Number of CT cycles required: two, *n* (%)	21 (20.2)	21 (16.5)	
Number of CT cycles required: three, *n* (%)	54 (51.9)	66 (52.0)	
Number of CT cycles required: four or more, *n* (%)	19 (18.3)	15 (11.8)	
Most common CT delivered: alkylating agents, *n* (%) ^4^	80 (61.5)	69 (46.6)	0.013 ^6^
Most common CT delivered: anthracyclines, *n* (%) ^4^	80 (61.5)	64 (43.2)	0.002 ^6^
Most common CT delivered: antimetabolites, *n* (%) ^4^	35 (26.9)	59 (39.9)	0.023 ^6^
Most common CT delivered: taxanes, *n* (%) ^4^	56 (43.1)	50 (33.8)	0.111 ^6^
Most common CT delivered: aromatase inhibitors, *n* (%) ^4^	63 (49.6)	39 (27.3)	<0.001 ^6^
Most common CT delivered: tamoxifen, *n* (%) ^4^	56 (43.4)	47 (32.4)	0.061 ^6^
Most common CT delivered: platinum analogues, *n* (%) ^4^	12 (9.2)	36 (24.3)	0.001 ^6^
Patients who received RT, *n* (%)	99 (76.2)	91 (61.5)	0.009 ^6^
Number of RT sessions required: one, *n* (%)	93 (93.9)	85 (93.4)	0.880 ^6^
Number of RT sessions required: two or more, *n* (%)	6 (6.1)	6 (6.6)	

Abbreviations: ca.: carcinoma, CT: chemotherapy, duct.: ductal, inv.: invasive, lob.: lobular, No.: number, RT: radiotherapy. Relative frequencies were calculated for the population without missing values. ^1^ Present in at least three patients in either cohort, coded according to MedDRA. ^2^ These are all subtypes of colon cancer that appear separate because they are coded under different terms according to MedDRA. ^3^ These are all subtypes of breast cancer that appear separate because they are coded under different terms according to MedDRA. ^4^ Present in at least 10% of patients in either cohort. These agents were as frequent among patients with possible neuropathic pain—as per the DN4—as among patients with unlikely neuropathic pain (63% vs. 67%, respectively, for alkylating agents; 65% vs. 60% for antitumor antibiotics; 69% vs. 62% for antimetabolites; 64% vs. 65% for taxanes; 47% vs. 53% for aromatase inhibitors; and 67% vs. 64% for platinum analogues). ^5^ Mann–Whitney U test. ^6^ Pearson’s chi-square test. ^7^ Fisher’s exact test.

**Table 3 cancers-16-01581-t003:** Clinical endpoints.

	With Pain (*n* = 130)	Without Pain (*n* = 148)	*p*-Value
Patients with positive (≥4) DN4 scores (*n* = 130, 100%), *n* (%)	76 (58.9)	NA	-
EQ5D: Quality of Life Index (*n* = 276, 99.3%), median (IQR) ^1^	0.6 (0.4)	1 (0.2)	<0.001 ^7^
EQ5D: health status score (*n* = 268, 96.4%), median (IQR) ^2^	70 (30)	80 (20)	<0.001 ^7^
HADS: anxiety score (*n* = 278, 100%), median (IQR) ^3^	4.5 (8)	3 (4)	<0.001 ^7^
HADS: depression score (*n* = 277, 99.6%), median (IQR) ^3^	3 (7)	1 (2)	<0.001 ^7^
HADS: total score (*n* = 277, 99.6%), median (IQR) ^4^	9 (13)	4 (6)	<0.001 ^7^
PCS: total score (*n* = 269, 96.8%), median (IQR) ^5^	7 (18)	1 (2)	<0.001 ^7^
WPAI-GH: cur. paid employment (2022) (*n* = 278, 100%), *n* (%)	29 (22.3)	51 (34.5)	0.026 ^8^
WPAI-GH: presenteeism score (*n* = 80, 100%), median (IQR) ^6^	0 (0)	0 (0)	0.561 ^7^
WPAI-GH: disability score (*n* = 278, 100%), median (IQR) ^6^	0 (40)	0 (30)	0.029 ^7^
WPAI-Pain: cur. paid employment (2022) (*n* = 278, 100%), *n* (%)	32 (24.6)	55 (37.2)	0.024 ^8^
WPAI-Pain: disability score (*n* = 277, 99.6%), median (IQR)	50 (70)	0 (0)	<0.001 ^7^
Symptoms associated to pain: insomnia (*n* = 277, 99.6%), *n* (%)	84 (65.1)	56 (37.8)	<0.001 ^8^
Symptoms associated to pain: fatigue (*n* = 277, 99.6%), *n* (%)	75 (58.1)	56 (38.1)	0.001 ^8^

Abbreviations: cur.: current, DN4: *Douleur Neuropathique 4 Questions*, EQ5D: EuroQol 5 Dimensions, HADS: Hospital Anxiety and Depression Scale, IQR: interquartile range, PCS: Pain Catastrophising Scale, WPAI GH: General Health-related Work Productivity and Activity Impairment Scale, WPAI Pain: Pain-related Work Productivity and Activity Impairment Scale. Relative frequencies were calculated for the population without missing values. ^1^ Index is based on individual preferences, and the indices of preference values for each health state are obtained from general population or patient groups’ studies. The Quality of Life Index oscillates from one (best health state) to zero (death), although there may be negative values that correspond to those health states valued as worse than death. ^2^ Scored from zero (worst health status possible) to 100 (best health status possible). ^3^ Scored from zero (no anxiety or depression) to 21 (maximal anxiety or depression), with scores ≥10 suggesting pathology. ^4^ Sum of the anxiety and depression scores, ranging from 0 to 42. ^5^ Scored from 0 to 52, with higher scores indicating more catastrophism. ^6^ Scored from 0 (no presenteeism or disability) to 100 (complete presenteeism or disability). ^7^ Mann–Whitney U test. ^8^ Pearson’s chi-square test.

**Table 4 cancers-16-01581-t004:** Correlates of pain (multivariable logistic regression).

	Estimate ^1^	Lower 95% CI Limit ^1^	Upper 95% CI Limit ^1^	*p*-Value
Intercept	−1.26	−4.41	1.89	0.432
Age, change per additional year	−0.02	−0.05	0.01	0.243
Male gender	−2.60	−4.29	−0.92	0.002
BMI, change per 1-unit (kg/m^2^) increase	0.07	0.01	0.14	0.029
Being married	0.12	−0.48	0.73	0.688
Having superior studies	0.22	−0.44	0.89	0.513
Smoking or routine alcohol intake	0.08	−0.63	0.80	0.820
Having any chronic disease	1.43	0.78	2.08	<0.001
Time since cancer onset, chg. per year	0.02	−0.03	0.08	0.430
Having had breast cancer	−1.49	−2.89	−0.08	0.038
Having had advanced cancer	−0.51	−1.37	0.36	0.251
Two surgeries (vs. one or none)	0.49	−0.19	1.17	0.157
Three or more surgeries (vs. one or none)	1.91	0.81	3.01	0.001
One or two chemotherapies (vs. none)	−1.08	−2.30	0.15	0.085
Three or more chemotherapies (vs. none)	−1.45	−3.00	0.09	0.065
Radiotherapy	0.52	−0.21	1.24	0.161
Vinca alkaloids or platinum compounds	−0.09	−1.44	1.26	0.896
Taxanes	0.12	−0.63	0.86	0.755
Anthracyclines	0.72	−0.85	2.29	0.370
Nitrogen mustards	−0.01	−0.85	2.29	0.370
Aromatase inhibitors	0.75	0.04	1.46	0.038

Abbreviations: BMI: body mass index, chg.: change, Kg: kilogram, m: metre. ^1^ A logit link was used in this binary model; thus, the estimates are log odds ratios.

**Table 5 cancers-16-01581-t005:** Correlates of possible neuropathic (DN4 score ≥ 4) pain (multivariable logistic regression).

	Estimate ^1^	Lower 95% CI Limit ^1^	Upper 95% CI Limit ^1^	*p*-Value
Intercept	0.26	−4.33	4.87	0.909
Age, change per additional year	−0.05	−0.10	0.01	0.029
Male gender	−1.11	−4.21	2.00	0.489
BMI, change per 1-unit (kg/m^2^) increase	0.12	0.03	0.21	0.010
Being married	0.63	−0.30	1.56	0.185
Having any chronic disease	0.01	−1.14	1.17	0.987
Time since cancer onset, chg. per year	−0.03	−0.12	0.07	0.631
Having had breast cancer	−1.84	−4.04	0.36	0.101
Having had advanced cancer	0.47	−0.90	1.84	0.501
Two surgeries (vs. one or none)	0.95	−0.20	2.09	0.105
Three or more surgeries (vs. one or none)	1.51	<0.01	3.01	0.049
One or two chemotherapies (vs. none)	1.22	−0.81	3.24	0.239
Three or more chemotherapies (vs. none)	1.69	−0.82	4.20	0.187
Radiotherapy	0.85	−0.44	2.13	0.198
Vinca alkaloids or platinum compounds	−0.97	−3.14	1.20	0.381
Taxanes	−0.33	−1.49	0.83	0.578
Anthracyclines	0.59	−2.31	3.49	0.692
Nitrogen mustards	−1.70	−4.77	1.36	0.277
Aromatase inhibitors	0.16	−0.90	1.23	0.757

Abbreviations: BMI: body mass index, chg.: change, Kg: kilogram, m: metre. ^1^ A logit link was used in this binary model; thus, the estimates are log odds ratios.

## Data Availability

Dr. Concepción Pérez, the Head of the Pain Clinic of the Hospital de la Princesa, will oversee the dataset. Granting access to this information will be evaluated on a case-by-case basis, upon reasonable request by the interested party. Data access requests should be addressed to Dr. Concepción Pérez at concha.phte@gmail.com.

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
