# Peer review of "Pain in Long-Term Cancer Survivors: Prevalence and Impact in a Cohort Composed Mostly of Breast Cancer Survivors"

_cancers, 2024, doi:10.3390/cancers16081581_

Round 1

Reviewer 1 Report

Comments and Suggestions for Authors

The manuscript represents an analysis of the incidence of neuropathic pain in cancer patients. it is clinically relevant issue, although the report does not have  novel outcomes. There are some important elements that have to be addressed. Essentially the group is formed by patients with ductal breast carcinoma which by themselves should be analysed the comparison with sigmoidal or colorectal is not correct based on therapy applied or surgery. The neuropathic pain recorded by the patient is similar, but the origin or the response to new therapy may differ, see  

  • DOI: 10.1016/j.jpainsymman.2014.11.300
  •  Table 4 male gender does not make sense as a point of comparison with other factors. Another element is BMI which is generally incorrect marker of analysis, water retention as well other elements play a role in these patients. The other issue is the discussion and conclusion, the discussion is vague and the conclusion does not represent a response to the hypothesis. The authors partially discuss the limitations, but it should be in a separate issue within the manuscript.
  • Please eliminate the Note in page 1 and check for several grammatical mistakes, easy to correct.
  •  
Comments on the Quality of English Language

Minor grammatical mistakes were encountered.

Author Response

Question #1: Essentially, the group is formed by patients with ductal breast carcinoma which by themselves should be analyzed the comparison with sigmoidal or colorectal is not correct based on therapy applied or surgery. The neuropathic pain recorded by the patient is similar, but the origin or the response to new therapy may differ, see DOI: 10.1016/j.jpainsymman.2014.11.300

Answer#1: We appreciate this objection since it is perhaps the weakest aspect of this research. In response to it, we have made a number of modifications that we summarize next:

1) We have included the term “breast” in the title (see also our response to another referee’s comments).

2) We have softened the discussion and conclusions, and have further elaborated this issue among the limitations (idem).

3) We have included the recommended citation in the discussion, but note that these and other authors have found that the impact of neuropathic pain in cancer survivors remains somewhat constant across different cancer types, which reinforces the transposability of our results to other subpopulations.

Question #2: Table 4 male gender does not make sense as a point of comparison with other factors. Another element is BMI which is generally incorrect marker of analysis, water retention as well as other elements play a role in these patients.

Answer #2: 

Please, note that we tried to be as exhaustive as possible in the adjusted analyses by including as many patients’ features that might act as confounders as possible. This is of particular relevance for the advanced analyses that we cite in the present manuscript and plan to convey in a separate one.

Although fluid retention might be of relevance while the oncologic disease or treatments are ongoing, this seems to be much less of a problem in cancer survivors once they have overcome the disease.

Question #3: The other issue is the discussion and conclusion, the discussion is vague and the conclusion does not represent a response to the hypothesis. The authors discuss the limitations, but it should be in a separate issue within the manuscript.

Answer #3: In response to your comment, we have abridged the discussion to focus on our particular findings. However, we think that the ideas we have left are of relevance for the readers because they contain a brief review of current therapeutic guidelines and provide some recommendations for personalizing them for cancer survivors.

Regarding the limitations, please see the response we have provided for your first comment.

Question #4: Please, eliminate the note in page 1 and check for several grammatical mistakes, easy to correct.

Answer #4: Following your request, we have removed the note on page 1 and had the manuscript re-reviewed by Sarah Cousins, a native English teacher with experience in copy-editing biomedical manuscripts.

Reviewer 2 Report

Comments and Suggestions for Authors

Thank you for the opportunity to review this manuscript, it is on an interesting topic. 

I have some minor concerns highlighted bellow. 

General comments. 

Please, change the “P value” to “p value” in the whole text. 

Specific comments

2.3. Clinical Endpoints

Could you add the scale used to measure the pain intensity? 

Could you explain a little more about the scales used? For example, the dimensions, the cut-off or the clinical significance of the score? For example, more punctuation, worse clinical situation? 

Line 126. Please add the software or formula used for the sample size estimation. 

Line 195. TNM stages. Please, explain the acronym and the clinical meaning of the scale. 

Line 371. “to pain specialists”, is it possible to add the next detail, please: “to pain specialists, including the use of non-pharmacological approaches, such us physiotherapy”. 

Some recent evidence supports the issue: 

Seth NH, Qureshi I. Effectiveness of physiotherapy interventions on improving quality of life, total neuropathy score, strength and reducing pain in cancer survivors suffering from chemotherapy-induced peripheral neuropathy - a systematic review. Acta Oncol. 2023 Sep;62(9):1143-1151. doi: 10.1080/0284186X.2023.2238890. Epub 2023 Jul 31. PMID: 37522184.”

“Pinheiro da Silva F, Moreira GM, Zomkowski K, Amaral de Noronha M, Flores Sperandio F. Manual Therapy as Treatment for Chronic Musculoskeletal Pain in Female Breast Cancer Survivors: A Systematic Review and Meta-Analysis. J Manipulative Physiol Ther. 2019 Sep;42(7):503-513. doi: 10.1016/j.jmpt.2018.12.007. PMID: 31864435.”

“Lahousse A, Reynebeau I, Nijs J, Beckwée D, van Wilgen P, Fernández-de-Las-Peñas C, Mostaqim K, Roose E, Leysen L. The effect of psychologically informed practice with behavioural graded activity in cancer survivors: systematic review and meta-analysis. J Cancer Surviv. 2023 Jan 26:1–46. doi: 10.1007/s11764-022-01270-4. Epub ahead of print. PMID: 36701101; PMCID: PMC9878499.”

Author Response

Question #1: Please, change the “P value” to “p value” in the whole text.

Answer #1: Done as recommended.

Question #2: 2.3 Clinical endpoints: Could you add the scale used to measure the pain intensity?

Answer #2: Thank you for your suggestion. We had not realized that we had not provided sufficient detail on this respect. In fact, we used the Brief Pain Inventory for this purpose. Following your comment, we have included it on Section 2.3 of the revised manuscript.

Question #3: 2.3 Clinical endpoints: Could you explain a little more about the scales used? For example, the dimensions, the cut-off or the clinical significance of the score? For example, more punctuation, worse clinical situation?

Answer #3: Following your suggestion, we have included both, references for the measurement tools used, and details about scoring ranges and interpretation. Please, note that due to space constraints, we have referred the readers to the citations provided for more technical details, and have provided the scoring ranges as footnotes of Table 3.

Question #4: Line 126. Please, add the software or the formula used for sample size estimations.

Answer #4: Done as recommended.

Question #5: Line 195. TNM stages. Please, explain the acronym and the clinical meaning of the scale.

Answer #5: Following your comment, we have included, in the revised manuscript, an URL of the National Cancer Institute where the readers can find definitions and more details about the TNM cancer staging.

Question #6: Line 371. “[…] to pain specialists”, is it possible to add the next detail, please: “to pain specialists, including the use of non-pharmacological approaches, such as physiotherapy”. Some recent evidence supports the issue: “Seth NH, Qureshi I. Acta Oncol 2023;62:1143-51”, “Pinheiro da Silva F, Moreira GM, Zomkowski K, Amaral de Noronha M, Flores Sperandio F. J Manipulative Physiol Ther 2019;42:503-13”, “Lahousse A, Reynebeau I, Nijs J, Beckwée D, van Wilgen P, Fernández-de-Las-Peñas C, Mostaqim K, Roose E, Leysen L. J Cancer Surviv 2023 Jan 26:1-46”.

Answer #6: Thanks for the contributions regarding the effects of physiotherapy in cancer survivors. Please, note that rather than including the proposed references, we have reorganized this paragraph in the discussion to include the entire spectrum of therapies that pain clinics can provide, in addition to physiotherapy.

Reviewer 3 Report

Comments and Suggestions for Authors

Thank you for permitting me to review this manuscript 

In this single center retrospective analysis the authors described the incidence of pain in  patients having a cancer in a 5 year period ,   multivariables regression analyses  was performed to better detect influence of other factor on pain in this population of patients 

Here are my suggestions 

please elaborate the specifications of this hospital with regard to cancer treatement ,  may be the percentage of cancer patients compared to other non cancer patients 

the large majority of patients are brease cancer patients , I suggest to remove other patients and focus only on these patients , and also add breast in the title 

conclusion should be much more cautious because of the shorthcomings cited  just before 

Author Response

Question #1: Please, elaborate the specifications of this hospital with regard to cancer treatment, may be the percentage of cancer patients compared to other non-cancer patients?

Answer #1: We appreciate your suggestion since it represents a worthy contribution. Fortunately, we had data on the numbers of patients attending the outpatient consultations, including the Medical Oncology Department, and the prevalence of pain in this setting, which we have included in the revised manuscript.

Question #2: The large majority of patients are breast cancer patients. I suggest to remove the other patients and focus only on these patients, and also add breast in the title. Conclusion should be much more cautious because of the shortcomings cited just before.

Answer #2: We acknowledge your suggestion to be more restrained in the discussion and conclusions given that out cohort was composed mostly of breast cancer survivors. Although, we do not deem it necessary to remove the other patients since the study protocol was comprehensive of all types of cancer, and the sample composition was determined by the casual diagnoses of presenting patients, we have modified the title and the conclusions, and placed this fact first among the limitations. Please, note that we have also elaborated this issue a bit more in the latter section.

Round 2

Reviewer 1 Report

Comments and Suggestions for Authors

The authors have responded to the critical questions raised. I consider it suitable for publication

Comments on the Quality of English Language

Several minor grammatical mistakes were encountered.

Reviewer 3 Report

Comments and Suggestions for Authors

The authors have adequately reponded to my queries